# Chronic Toxicity of Primary Metabolites of Chloroacetamide and Glyphosate to Early Life Stages of Marbled Crayfish *Procambarus* *virginalis*

**DOI:** 10.3390/biology11060927

**Published:** 2022-06-17

**Authors:** Nikola Tresnakova, Jan Kubec, Alzbeta Stara, Eliska Zuskova, Caterina Faggio, Antonin Kouba, Josef Velisek

**Affiliations:** 1South Bohemian Research Center of Aquaculture and Biodiversity of Hydrocenoses, Research Institute of Fish Culture and Hydrobiology, Faculty of Fisheries and Protection of Waters, University of South Bohemia in Ceske Budejovice, Zatisi 728/II, 389 25 Vodnany, Czech Republic; tresnakova@frov.jcu.cz (N.T.); kubecj@frov.jcu.cz (J.K.); staraa01@frov.jcu.cz (A.S.); zuskova@frov.jcu.cz (E.Z.); akouba@frov.jcu.cz (A.K.); velisek@frov.jcu.cz (J.V.); 2Department of Chemical, Biological, Pharmaceutical and Environmental Sciences, University of Messina, Viale Ferdinando Stagno d’Alcontres 31, 98166 Messina, Italy

**Keywords:** metabolite, herbicide, behaviour, ontogeny, toxicity, crayfish

## Abstract

**Simple Summary:**

Due to the rising population, it is necessary to ensure sustainable agricultural management, especially in agricultural production. Therefore, using pesticides is the only way to reach a sufficient number of crops for the human population. However, using these agrochemicals has one big disadvantage—harmful effects on non-target species. Since aquatic ecosystems are essential for biota, fauna, flora, and humans, this study provides information about the toxicity of primary metabolites of commonly used glyphosate and acetochlor herbicides. Generally, it is known that pesticide degradation products may have lower, the same, or even higher toxicity than their parental compounds. Moreover, the developmental stages are more predisposed to be affected by pollution than adults, and consequently, in the worst case, it may affect the whole species population. Due to this reason, we studied the toxicity of these metabolites on embryolarval marbled crayfish—how they can affect their growth, ontogenetic development, behaviour, and gill morphology. Our study revealed that these two metabolites alone induce changes in behaviour, antioxidant enzymes, and gill morphology, even in combination. Nevertheless, there is a gap in the knowledge about pesticides’ metabolite toxicities, a small fraction of the hundreds with potential for entering the aquatic ecosystem. Therefore, chronic toxicity tests are suitable for evaluating the toxicity of aquatic pollution.

**Abstract:**

Degradation products of herbicides, alone and in combination, may affect non-target aquatic organisms via leaching or runoff from the soil. The effects of 50-day exposure of primary metabolites of chloroacetamide herbicide, acetochlor ESA (AE; 4 µg/L), and glyphosate, aminomethylphosphonic acid (AMPA; 4 µg/L), and their combination (AMPA + AE; 4 + 4 µg/L) on mortality, growth, oxidative stress, antioxidant response, behaviour, and gill histology of early life stages of marbled crayfish (*Procambarus virginalis)* were investigated. While no treatment effects were observed on cumulative mortality or early ontogeny, growth was significantly lower in all exposed groups compared with the control group. Significant superoxide dismutase activity was observed in exposure groups, and significantly higher glutathione S-transferase activity only in the AMPA + AE group. The gill epithelium in AMPA + AE-exposed crayfish showed swelling as well as numerous unidentified fragments in interlamellar space. Velocity and distance moved in crayfish exposed to metabolites did not differ from controls, but increased activity was observed in the AMPA and AE groups. The study reveals the potential risks of glyphosate and acetochlor herbicide usage through their primary metabolites in the early life stages of marbled crayfish.

## 1. Introduction

Industrial and agricultural technologies can be a source of ecotoxic pesticides and their metabolites occurring in residual concentrations in various environmental compartments [1,2,3]. This is related to the change in pesticide use pattern with the increased cultivation of energy crops, such as maize and rapeseed, rather than traditional crops, such as cereals [4]. Although pesticides are, to a certain extent, removed by biotic and abiotic transformation, a potential risk posed by degradation products remains [5,6], including aquatic ecosystems [2,7,8]. While governmental and non-governmental organizations provide information and regulations on the use and effects of pesticides, gaps exist as to their global distribution and impact. Residual pesticides and their metabolite concentration have been detected in Europe and the United States at ng/L to low µg/L [6], and greater than 10 mg/L concentrations [9] lead to ecotoxicological concerns. When chemicals occur in combination, the toxicity of their metabolites may equal or exceed that of the parent compounds [10,11].

Acetochlor ethane sulfonic acid (ESA) is a primary metabolite of mutagenic chloroacetamide herbicide. Based on evidence showing reproductive and developmental impacts [12], acetochlor was banned in the Czech Republic in 2014 [13]. It is still applied on crop fields [14] in the USA and China to control grasses and selected broadleaf weeds [15]. Residual concentrations of parental compound acetochlor occurring in USA waters ranged from 2.11 [16] to 6.08 μg/L [17], and in China, from 0.0179 to 1.0549 μg/L [18,19]. However, its metabolite ESA is easily soluble in water, and due to leaching from the soil, it is commonly detected more frequently and even with higher concentrations [20,21,22,23,24,25,26,27,28,29,30]. More recent detailed information about detecting ESA in waters of countries still acetochlor is missing in the literature. However, in the Czech Republic, residual concentrations of ESA are detected in drinking water in a range of 0.14 to 0.24 μg/L [31]. Next, in surface waters, a maximum concentration in spring of 0.4 μg/L and autumn of 0.36 μg/L has previously been recorded [32]. The Kodeš [33] reported a maximal concentration of 4.19 μg/L in the waters of the Czech Republic.

The residual concentrations of acetochlor and its metabolite occurrence in aquatic ecosystems may induce adverse effects in non-target aquatic organisms. For example, acetochlor affects thyroid hormone levels [34,35,36], oxidative stress [37,38], antioxidant enzymes levels [39,40], histology [39,41], early development [34,35,37,39,42], immune system [42], and behaviour [38,41]. Moreover, acetochlor can be accumulated in organisms, posing a risk to the entire ecosystem through the food web [40]. Compared with acetochlor, there are minimal data about the potential health effects of ESA. According to the Minnesota Department of Health [43], short-term and longer-term animal studies showed that exposure to ESA led to changes in thyroid hormones, and high doses affected the male reproductive system. Longer-term animal studies also showed changes in overall body weight. Based on animal studies, acetochlor ESA appears less potent than the parent compound acetochlor. Further, the U.S. EPA [44] considers acetochlor metabolites unlikely to be carcinogenic and significantly less toxic.

Aminomethylphosphonic acid (AMPA) is a significant metabolite of glyphosate-based herbicides (GBH), belonging to a chemical group called aminomethylenephosphonates. In addition to GBHs, sources of AMPA in the environment include chemicals used in water treatment [45,46], detergents, and chemicals used in industrial boilers and cooling devices. Glyphosate-based herbicides are broad-spectrum systematic herbicides and crop desiccants, primarily used on cereal grains, maize, and rapeseed [46,47]. Aminomethylphosphonic acid, with a half-life of 76 to 240 days, shows the maternal glyphosate’s toxicity three- to six-fold [48]. The presence of AMPA is commonly reported in freshwaters, sediment, and suspended particulates [45,49], more frequently than in glyphosates [47,49,50]. Maximum reported concentrations of AMPA were 44 μg/L in the Lauch River (France), 5.7 μg/L in the south fork of the Iowa River (Iowa, USA), and 2.6 μg/L in the Bogue Phalia River (Mississippi, USA) [51]. Other studies from the US report a maximum concentration of AMPA in vernal pools above 3 μg/L, in streams of 0.22 and 3.6 μg/L, and treated wastewater of 3.9 μg/L [49,50,52,53].

Concentrations used in several studies were usually 100- to 1000-times the predicted environmental surface water at low concentrations of AMPA [49]. Nonetheless, exposure to AMPA may cause physiological changes in aquatic invertebrate species, fish, and amphibians [54]. For example, an acute toxicity value was established for guppy *Poecilia reticulata* of 180 mg/L [55]. Chronic exposure to AMPA of fathead minnow *Pimephales promelas* and *Daphnia magna* were studied by Levine et al. [46]. They reported a no-observed-effect concentration (NOEC) of 12 mg/L for fathead minnow and 15 mg/L for *D*. *magna*.

It is generally known that metabolite toxicities may be similar to or even higher than their parental compounds. As mentioned above, several toxicity studies have already investigated the individual effects of AMPA and acetochlor. Nevertheless, the mixture of AMPA and ESA, which are commonly presented in the environment, and their parental compound [33,49,56,57,58,59] effects on aquatic non-target organisms are still unknown. Therefore, it is necessary to investigate their toxicity to non-target aquatic organisms under similar exposure conditions. Crayfish are one of the most environmentally significant species in aquatic ecosystems and the food web. Further, these species play an essential role as bioindicators in toxicology studies of water pollution [60]. The metabolism of many xenobiotics is slower in invertebrates than in fish [61,62], and, consequently, invertebrates retain higher contaminant residue levels [63]. Currently, information on the toxicity of these metabolites to early life stages of crayfish is scant, both of short-term acute impact as well as their chronic effects, potentially of greater concern. Ecotoxicity analyses using appropriate biomarkers and indicators to identify the effects of toxic substances in natural populations are needed.

This study aimed to assess the effects of acetochlor ESA and AMPA, and their combination, at environmental concentrations on early life stages of marbled crayfish *Procambarus virginalis,* concerning (1) mortality, (2) growth, (3) ontogenetic development, (4) behaviour, (5) oxidative stress response and antioxidant biomarkers, and (6) histology.

## 2. Materials and Methods

### 2.1. Test Substances

The tested metabolites were purchased from Sigma-Aldrich Corporation (St. Louis, MO, USA): acetochlor ESA (chemical purity 98.6%; CAS: 947601-84-5) and AMPA (chemical purity 98.5%, CAS: 1066-51-9).

Before and after solution exchange, liquid chromatography-electrospray ionization-mass spectrometry was used to check ESA and AMPA concentrations according to Aparicio et al. [64] and Yokley et al. [65], respectively. Measured values did not differ from the nominal concentration by more than ±10% for acetochlor ESA and ±13% for AMPA.

### 2.2. Test Organism

Juveniles from a single parthenogenetically reproducing marbled crayfish (*Procambarus virginalis*) female (carapace length 49.5 mm, and weight 26.6 g) were used. The parent crayfish originated from the laboratory culture of the Faculty of Fisheries and Protection of Waters at the University of South Bohemia in Ceske Budejovice (Czech Republic).

### 2.3. Experimental Protocol

First, 288 stage three juveniles (mean weight 6.81 mg) were divided into three experimental groups and a control group (each n = 36/group) and subjected to exposures as follows:AE: 4.0 µg/L acetochlor ESAAMPA: 4.0 µg/L aminomethylphosphonic acidAE + AMPA: (4.0 µg/L AE + 4.0 µg/L AMPA)Control just 20 mL tap water only.

Chemicals were used at concentrations reported in Czech waters by Kodeš [33], considering published concentrations previously mentioned. All treatments were performed in duplicate.

Crayfish were individually placed into three macroplates (12 crayfish per each) with 35 mL wells per group with a light regime of 12 h:12 h light:dark. Exchanging of water was maintained thrice per week by gentle water draining from each well, and then a new solution was added.

During the experiment, water quality parameters (t 20.8–21.2 °C, DO > 95%, pH 7.75–8.03) were measured daily by multimeter SI Analytics ProLab 2000 (SCHOTT Instruments, Germany) and Minikin loggers (Environmental Measuring Systems, Brno, Czech Republic). Daily mortality and moulting were noticed, as well as providing feed *ad libidum* (freshly hatched *Artemia* sp. nauplii) to crayfish. Dead specimens and crayfish’s ecdysis were removed to avoid organic pollution of waters. Due to safe manipulation, each developmental stage of individual crayfish was measured on the second day after moulting (reaching partial calcification of the exoskeleton). The test was terminated at 50 days. Surviving specimens were observed for behaviour patterns, after which they were weighed, killed by ice anaesthesia [66,67] and stored at −80 °C for further processing or histopathological observation.

### 2.4. Tested Parameters

#### 2.4.1. Growth Rate

Crayfish were weighed on a Mettler-Toledo (Greifensee, Switzerland) analytical balance two days after moulting and after 50 days. The mean specific growth rate (SGR) was calculated for the period from day five (M_5_, first samples taken) through day 50 (M_50_, end of the experiment). Data of experimental treatments were compared with those of controls according to OECD Guideline for testing of chemicals protocol no. 215 [68].

The inhibition of specific growth rate in each experimental group was calculated using the following formula according to OECD number 215 [68]:$$I_{x}\left[\%\right]=\frac{SGR_{x}(control)-SGR_{x}(group)}{SGR_{x}(control)}\cdot 100$$
where is $I_{x}$ = inhibition of specific growth in selected experimental group after x days of exposure, $SGR_{x}(control)$ = mean specific growth rate in the control group, and $SGR_{x}(group)$ = mean specific growth rate in selected experimental group.

#### 2.4.2. Early Ontogeny

Occurrence of body deformities and morphological anomalies, such as deformed pereiopods and walking legs, lacking the uropods, and degree of ontogenetic development, were assessed according to Vogt [69].

#### 2.4.3. Behaviour

The behaviour of surviving juveniles was evaluated following the protocol by Velisek et al. [11]. Briefly, crayfish were recorded in 50 mL of the specific bath according to experimental groups for 60 min using a digital video camera (Sony HDR-CX240, Sony, Japan) attached above arenas. Locomotion, including distance, moved (cm), activity (percentage of time locomotion was detected), and velocity (cm/s) were estimated with EthoVision^®^ XT 15 software (Noldus Information Technology, Wageningen, The Netherlands).

#### 2.4.4. Oxidative Stress and Antioxidant Biomarkers

For assessment of oxidative stress and antioxidant enzymes, at the end of the trial, the entire bodies of two crayfish in six repetitions from each group were homogenized and pooled and prepared for analysis following Stara et al. [70]. First, pooled samples were weighed, combined with phosphate-buffered saline (PBS, pH 7.2) at 1 mL PBS/100 mg tissue, and homogenized using a ball homogenizer (TissueLyserII, QIAGEN^®^). Except for lipid peroxidation (LPO), evaluated according to Lushchak et al. [71], the homogenates were centrifuged at 30,000× *g* at 4 °C for 30 min for parameters superoxide dismutase (SOD; EC 1.15.1.1) and catalase (CAT; EC 1.11.1.6), and at 10,000× *g* for 15 min at 4 °C for glutathione S-transferase (GST; EC 2.5.1.18) and acetylcholinesterase (AChE; EC 3.1.1.7.) following protocols by Marklund and Marklund [72], Beers and Sizer [73], Habig et al. [74], and Ellman et al. [75]. Every sample was spectrophotometrically measured in three repetitions with plate reader M200 (Switzerland). Prior to the enzyme analyses, protein levels were determined according to Bradford [76], and biomarkers are expressed in international units (miliunits) per mg of protein.

#### 2.4.5. Histology

The six crayfish from each group were routinely processed for histology and interpreted according to Ceccaldi [77]. Briefly, tissues were fixed in 10% buffered formalin for 24 h and decalcified for 4 h (slow decalcifier DC1; containing formic acid and formaldehyde, Labonord SAS, Templemars, France), processed in Histomaster 2052/1,5, and embedded in paraffin. Five μm sections were cut on a rotary microtome (LS-2065A) and stained with haematoxylin-eosin (Tissue-Tek^®^ DRS™ 2000, Sekura, Finetek U.S.A.). Hepatopancreas and gill were examined by light microscopy and photographed (E-600 Olympus BX51, Tokyo, Japan). Sections were scored as (0) no pathology; (1) pathology in ≤20% of the fields; (2) pathology in 20–60% of the fields; and (3) pathology in >60% of the fields.

### 2.5. Statistical Analysis

Contingency tables (χ^2^) analysis was used for cumulative mortality data. Normality and homoscedasticity of variance were checked with the Kolmogorov–Smirnov test and Bartlett’s test, respectively. The differences in measured variables among groups were evaluated with a one-way ANOVA if conditions were satisfied. After difference detection (*p* < 0.05), the Tukey Unequal N HSD test was used. The histopathological score (0–3) was determined for each crayfish and organs examined and statistically analysed using a nonparametric Kruskal–Wallis test using Statistica v. 12.0 for Windows (StatSoft, version 12.0).

## 3. Results

### 3.1. Cumulative Mortality and Growth

No significant differences (H = 2.61, *p* = 0.231) among groups were found in cumulative mortality. Mortality was 16.7 ± 1.0% in the control, 13.9 ± 1.20% in AE, 11.1 ± 0.90% in AMPA, and 19.4 ± 0.80% in AE + AMPA.

From developmental stage V, the exposed groups showed significantly (H = 9.49, *p* = 0.008) lower weight compared with the control (Figure 1). Specific growth rates are given in Table 1. Inhibition of growth with exposure to AE, AMPA, and AE + AMPA was 9.13, 10.96, and 13.93% compared to control, respectively.

### 3.2. Early Ontogeny

Exposure to AE, AMPA, and AE + AMPA showed no significant ((H = 4.13, *p* = 0.221) effects on crayfish’s early ontogeny. After the trial, most crayfish in all groups reached stage VIII (Figure 2).

### 3.3. Behaviour

No significant differences were detected in pesticide-exposed crayfish groups compared with controls in the distance moved (H = 4.68, *p* = 0.196) or velocity (H = 6.93, *p* = 0.074) (Figure 3A,B). Crayfish exposed to AMPA and AE exhibited increased activity compared to control (H = 13.00, *p* = 0.005), while AE + AMPA was not significantly different (Figure 3C).

### 3.4. Oxidative Stress and Antioxidant Response

Evidence of the effects of AE, AMPA, and AE + AMPA on oxidative stress and antioxidant responses in the whole-body homogenate of juvenile crayfish is presented in Table 2. Crayfish in all exposed groups showed significantly (H = 9.71 *p* = 0.006) higher SOD activity compared to controls, and those exposed to AE + AMPA showed significantly ((H = 9.68, *p* = 0.007) higher GST compared to controls. No significant differences among groups were observed in TBARS level, CAT activity, or AChe value.

### 3.5. Histology

The gill morphology of control, AE, and AMPA groups was similar, while moderate pathology was apparent in the AE + AMPA group (histopathological score median = 2), consisting of focal swelling of the epithelium with numerous unidentified fragments in interlamellar space (Figure 4). Hepatopancreas tissue in exposed groups did not differ from control.

## 4. Discussion

Xenobiotic substances, including parent compounds and their metabolites in aquatic ecosystems, may cause developmental delay, malformations, behaviour changes, and mortality in non-target organisms [41,78,79]. Our study focused on crayfish, which play a crucial role in aquatic ecosystems in food web energy transport [80]. Assessment of the effects of chronic exposure, as opposed to acute, provides a greater range of information with respect to the impact of a given compound on non-target organisms. As the general assumption is that herbicide degradation products may have short- and long-term effects like their parent compounds [26], and data on the toxicity of AMPA and EA on aquatic invertebrates are minimal, the results of this study are discussed in light of the available published information of the tested metabolites, their parent compounds, and other herbicides.

As expected, according to available studies, no significant differences in cumulative mortality were detected among the tested groups. By contrast, the 24 and 96 h LC50 values of the herbicide acetochlor were published by Yu et al. [41] for red swamp crayfish (*Procambarus clarkii*) at 145.24 and 191.25 mg/L, respectively. It is reported to exert toxic effects on red swamp crayfish, including loss of balance, body sway, and lethargy [41]. AMPA was not shown to be acutely toxic to zebrafish *Danio rerio* embryos at concentrations of 1.7, 5, 10, 23, 50, or 100 mg/L in 50–96 h trials [81]. The cited authors reported AMPA toxicity like that of glyphosate. Fiorino et al. [82] observed cumulative mortality ≤ 10% after 96 h exposure of *Danio rerio* embryos to glyphosate at 50 mg/L. Lethal concentrations of the AE and AMPA metabolites were determined to be 100- to 1000-times that predicted in environmental surface water [12]. We found no reports of mortality induced by acetochlor ESA in the literature.

We observed the body weight of all exposed crayfish groups at the fifth developmental stage to be significantly lower than that of the control group. In contrast, Levine et al. [46], for 21 days of exposure to AMPA (0.75 mg/L, 1.5 mg/L, 3.0 mg/L, 6.0 mg/L, and 12), reported no effect on the growth of *D. magna* (NOEC 15 mg/L) or *P. promelas* embryos (NOEC 12 mg/L). Impacts on growth have been commonly observed in the early life stages of crayfish exposed to pesticides and their metabolites [11,41,83,84] as well as in the early life stages of fish [1,84]. These and our results imply that growth is a more sensitive parameter than mortality. Retarded growth could be explained by diverting energy to essential physiological and biochemical processes in tissues [42,85,86,87].

Early ontogenetic development is considered a sensitive biomarker for examining the impacts of pesticides on fish [42,87,88] and crayfish [89,90]. However, in our study, crayfish’s early ontogeny did not show potential as a biomarker for monitoring AMPA and AE metabolites in the aquatic environment, as no exposure showed adverse effects on the ontogeny of marbled crayfish. This was similar to observations of chronic exposure to chloridazon residues [11] and triazine metabolites [91] on marbled crayfish. The early life stages of crayfish are more susceptible to water pollution, nutritional deprivation, and predation than adults, in both natural and cultured conditions [69]. Xenobiotic impact on mortality, growth, and the early ontogeny of aquatic organisms may be species-dependent [55,87] and related to their combinations [41].

The study follows previous research [89,90] exploring pesticide-specific effects on the behaviour of marbled crayfish as a model invertebrate under defined consistent conditions. The observations confirm that low concentrations of herbicides can have considerable consequences. Significant differences in AMPA- and AE-exposed groups from control and AE + AMPA crayfish activity suggest the potential of both metabolites to change locomotor behaviour. These alterations could affect energy use, leading to a shorter life span in more active individuals [92]. More active animals are also at increased predation risk, especially a factor in invaded ecosystems. The higher activity of non-native crayfish, usually associated with altered foraging behaviour, may place food resources under significantly more pressure [93] and alter ecosystem functioning [94]. The elevated activity can lead to the breakdown of food webs, biodiversity loss, and ecosystem instability [95,96].

Oxidative stress biomarkers are widely used for monitoring the aquatic environment [97,98]. The responses of an organism to xenobiotics may depend on its ability to neutralize reactive oxygen species (ROS) [70,99]. A high level of ROS in tissue may constitute a predisposition to intracellular damage because of membrane lipid peroxidation as well as DNA and protein injury, resulting in oxidative stress and, potentially, mortality [100,101]. Crayfish exposed to AMPA, AE, and their combination showed significantly (*p* < 0.01) higher SOD activity than did the control group. Those exposed to the variety showed significantly (*p* < 0.01) higher GST levels compared to controls. We did not find lipid peroxidation levels indicative of oxidative damage in any exposure group, suggesting that SOD and GST are the primary detoxifying antioxidants of AMPA and AE in the early life stages of crayfish, contributing to maintaining oxidative balance and preventing lipid peroxidation damage. Changes in antioxidant defence and maintenance of cell balance without effects on lipid peroxidation levels have been described in the homogenate of crayfish exposed to pesticides [11,90,102]. In other studies [70,103,104,105,106], increased TBARS levels usually indicated oxidative stress. These studies found oxidative damage in individual tissues of fish and crayfish, most commonly in the liver/hepatopancreas. It is the main organ of absorption, digestion, and storage of nutrients and is the main organ of xenobiotic detoxification, while oxidative damage to muscle after pesticide exposure is rarely found [70].

The gills of crayfish play an essential role in the transport of respiratory gases, nitrogen compound excretion, and osmoregulation. Gills are the initial barriers to waterborne pollutants. We observed swelling in the gill epithelium in groups exposed to AE + AMPA. These changes can result in reduced oxygen intake and disruption of osmoregulatory function. A pronounced decrease in oxygen consumption may lead to internal hypoxia, with effects on metabolism and locomotion [107]. In a persistent hypoxic state, many crustaceans adapt movements to reduce the oxygen consumed by muscles, shifting the available oxygen to the metabolism [108]. No histopathological changes were found in hepatopancreas tissues, suggesting that an antioxidant defensive system based mainly on enzymatic components, such as superoxide dismutase, catalase, and glutathione S-transferase sufficiently protects the cells themselves from ROS-induced cellular damage [109].

## 5. Conclusions

Exposure to the metabolites acetochlor ESA and AMPA, alone and in combination, can reduce growth but increase antioxidant enzyme levels in the early developmental stages of marbled crayfish. Combined exposure of acetochlor ESA and AMPA cause focal swelling in the gill epithelium with cell fragments in the interlamellar space in marbled crayfish.

This study examined only two herbicide metabolites, a small fraction of the hundreds with potential for entering the aquatic ecosystem, the toxic effects of which can be potentiated with combined exposures.

The study of herbicide metabolite toxicity is helpful in screening for chronic impacts of single and combined metabolites on early life stages of crayfish as a model non-target organism. Generally, lethal effects are only seen with pesticides at high levels. However, studying the long-term impacts of real environmental concentrations provides more pertinent information. Therefore, chronic toxicity tests should be a prerequisite when evaluating xenobiotics’ environmental effects on aquatic pollution.

## Figures and Tables

**Figure 1 biology-11-00927-f001:**
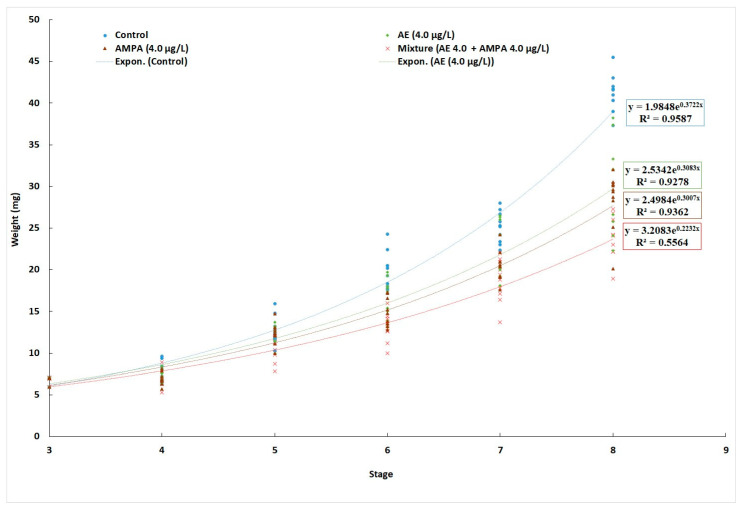
Body weight of early life stages of marbled crayfish *Procambarus virginalis* during 50-day exposure to acetochlor ESA (AE; 4.0 µg/L; N = 31), aminomethylphosphonic acid (AMPA; 4.0 µg/L; N = 32) and their combination (AE; 4.0 µg/L + AMPA; 4.0 µg/L; N = 29) and in control (no exposure, N = 30).

**Figure 2 biology-11-00927-f002:**
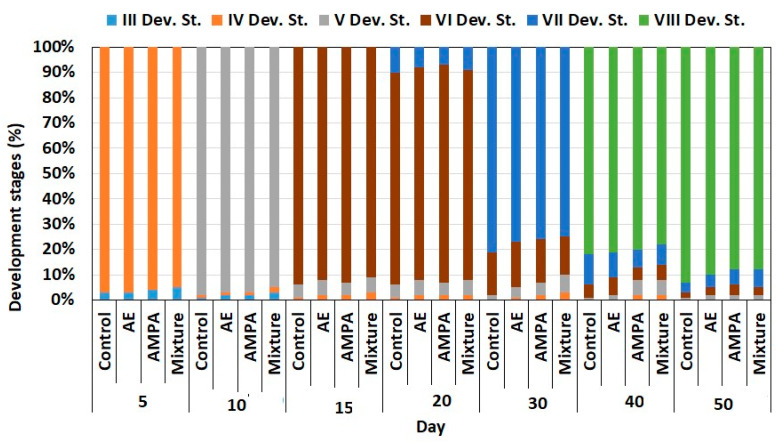
Marbled crayfish *Procambarus virginalis* developmental stages during 50-day exposure to acetochlor ESA (AE, 4.0 µg/L, N = 31), aminomethylphosphonic acid (AMPA, 4.0 µg/L, N = 32), and their combination (AE, 4.0 µg/L + AMPA, 4.0 µg/L, N = 29), and in control (no exposure, N = 30.

**Figure 3 biology-11-00927-f003:**
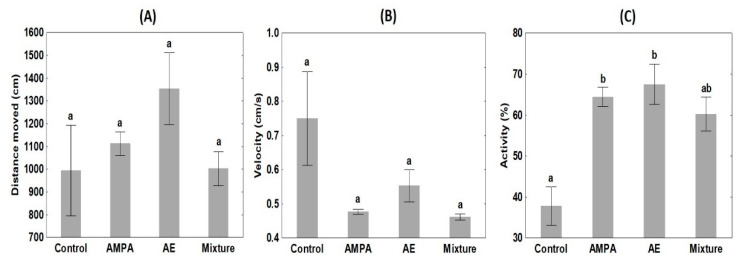
Distance moved (**A**), velocity (**B**), and activity (**C**) of the marbled crayfish *Procambarus virginalis* in untreated controls (N = 30) and groups exposed to acetochlor ESA (AE, 4.0 µg/L, N = 31), aminomethylphosphonic acid (AMPA, 4.0 µg/L, N = 32) and their combination (AE, 4.0 µg/L + AMPA, 4.0 µg/L, N = 29); different superscripts indicate significant difference among groups (*p* < 0.01). Data are means ± S.D.

**Figure 4 biology-11-00927-f004:**
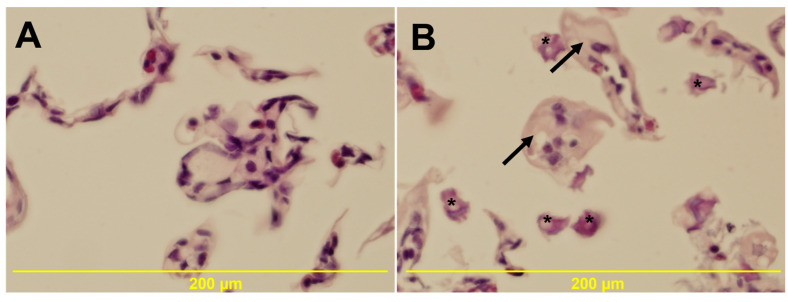
Transverse and longitudinal section of juvenile marbled crayfish *Procambarus virginalis* gill (N = 6; 200×). (**A**) gill of control crayfish; (**B**) gill of crayfish exposed to AE, 4.0 µg/L + AMPA, 4.0 µg/L for 50 days, with the most pronounced changes represented by swelling of gill epithelium (arrows) and unidentified fragments in interlamellar space (asterisk)—histopathological score 2.

**Table 1 biology-11-00927-t001:** Body weight of early life stages of marbled crayfish *Procambarus virginalis* during a 50-day exposure to acetochlor ESA (AE, N = 31) at 4.0 µg/L, aminomethylphosphonic acid (AMPA, N = 32) at 4.0 µg/L and their combination (AE + AMPA, N = 29) each at 4.0 µg/L, and in control (no exposure, N = 30); different superscripts indicate significant difference among groups (*p* < 0.01).

Treatment	Control (0 µg/L)	AE (4.0 µg/L)	AMPA (4.0 µg/L)	AE + AMPA (4.0 µg/L + 4.0 µg/L)
M_5_ (Mean ± SD), mg	7.94 ± 1.05 ^a^	7.55 ± 0.81 ^a^	7.66 ± 1.24 ^a^	7.40 ± 1.10 ^a^
M_50_ (Mean ± SD), mg	45.33 ± 3.52 ^a^	37.73 ± 7.96 ^b^	36.18 ± 4.50 ^b^	34.11 ± 8.74 ^b^
SGR	4.38 ± 0.08 ^a^	3.98 ± 0.10 ^b^	3.90 ± 0.09 ^b^	3.77 ± 0.12 ^b^

**Table 2 biology-11-00927-t002:** Antioxidant and oxidative biomarkers in whole-body homogenates of marbled crayfish *Procambarus virginalis* (N = 12) after 50-day exposure to acetochlor ESA (AE), aminomethylphosphonic acid (AMPA) and their combination (AE + AMPA); different superscripts indicate significant difference among groups (*p* < 0.01). Data are means ± S.D.

Treatment	Control (0 µg/L)	AE (4.0 µg/L)	AMPA (4.0 µg/L)	AE + AMPA (4.0 µg/L + 4.0 µg/L)
TBARS (nmol/mg protein)	0.214 ± 0.006 ^a^	0.246 ± 0.018 ^a^	0.247 ± 0.017 ^a^	0.219 ± 0.036 ^a^
SOD (nmol/min/mg protein)	0.060 ± 0.031 ^a^	0.199 ± 0.037 ^b^	0.205 ± 0.054 ^b^	0.136 ± 0.046 ^b^
CAT (µmol/min/mg protein)	0.326 ± 0.047 ^a^	0.406 ± 0.031 ^a^	0.424 ± 0.080 ^a^	0.387 ± 0.058 ^a^
GST (nmol/min/mg protein)	1.109 ± 0.038 ^a^	1.384 ± 0.229 ^ab^	1.416 ± 0.187 ^ab^	1.762 ± 0.576 ^b^
AChE (nmol/min/mg protein)	8.753 ± 1.921 ^a^	9.176 ± 4.658 ^a^	5.358 ± 1.794 ^a^	10.663 ± 3.270 ^a^

## Data Availability

The data in this study are available upon request from the first author.

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
