# Peer review of "Chronic Toxicity of Primary Metabolites of Chloroacetamide and Glyphosate to Early Life Stages of Marbled Crayfish Procambarus virginalis"

_biology, 2022, doi:10.3390/biology11060927_

Round 1
Reviewer 1 Report
This study reports the chronic toxicities of acetochlor ethanesulfonic acid, aminomethylphosphonic acid, and their combination on early life stages of marbled crayfish Procambarus virginalis. The determined endpoints including mortality, growth, oxidative stress, antioxidant response, behavior, and gill histology. The results are interesting and I think this manuscript merit publication in Biology, however some points remain to be revised or clarified before acceptance.
1. Simple Summary: The logic is confused, please rephrase it.
2. Line 33: change “Behaviour” to “behaviour”.
3. Figures 1 and 4: The resolution is low.
4. All the results are presented as mean ± standard error (S.E.) or mean ± standard deviations (S.D.)?
5. How many crayfish were used in each experiment (n=?)? Please provide the information in the figures and tables.
6. Figure 3: what’s the total activity? Please give more information in Materials and Methods section.
7. Line 337: change “locomotory Behaviour” to “locomotory behaviour.”.
Reviewer 2 Report
Review of the article: Chronic toxicities of acetochlor ethanesulfonic acid, ami-2 nomethylphosphonic acid, and their combination on early life 3 stages of marbled crayfish Procambarus virginalis
Manuscript ID: biology-1766677
In this study, the authors aimed to examine the effects of a 50-day exposure to acetochlor ESA and aminomethylphosphonic acid and their combination on mortality, growth, oxidative stress, antioxidant response, behaviour, and gill histology of early life stages of marbled crayfish. The work presents a good approach. Many information about the previous study findings is presented for readers to follow the present study rationale and procedures. I have only few comments on the manuscript.
1. The chemicals in this study were used at concentrations (4.0 μg/L) by referring to one reference (authored by Kodeš). Have you tried the chemicals at other concentrations and compared the differences of experimental results among them?
2. Only P values are presented in the manuscript for the statistical significance. The values of specific statistics (e.g., F values) and degree of freedom also need to be shown next to each P value.
3. At line 222, why not using nonparametric Kruskal-Wallis test to compare the data among more than two groups?
4. The graph in Figure 1 is poor in format (such as font size) and needs to be revised. The photos in Figure 4 are vague and should be replaced by others of better quality.
5. At line 238, the text should be revised.
6. In Table 1, no superscripts are shown for the values of SGR to indicate whether there is significant difference among groups.
7. All scientific names in the References should be italicized.
Reviewer 3 Report
Manuscript ID: biology-1766677
Title: Chronic toxicities of acetochlor ethanesulfonic acid, ami-nomethylphosphonic acid, and their combination on early life stages of marbled crayfish Procambarus virginalis
General Comments:
This manuscript examined the toxicity of the primary metabolites of 2 common agricultural pesticide to early life stages of marbled crayfish. The study is a good fit for the special issue and the choice of model organism adds novelty to the data presented. The intro had a lot of very good information and set up the manuscript well, however authors could expand on the importance of crayfish as a model organism. There are some major concerns that should be addressed prior to publication.
1. The grammar and writing in this manuscript is poor quality and difficult to read. Extensive editing is needed prior to publication (especially in the abstract).
2. Since all tests were performed with the offspring from one single crayfish, do the authors have any data to indicate that interindividual sensitivity between larval batches is low? How do we know that offspring from a single individual are representative?
Specific Comments:
Title: it may be more engaging for the reader if the authors instead wrote “Chronic toxicity of primary metabolites of chloracetamide and glyphosate to early life stages of marbled crayfish Procambarus virginalis”. It’s more succinct and clear.
Line 55 – Authors should include a brief description of the location these samples were measured (just country should suffice)
Line 139 – the acronym “FFPW USB” should be defined
Line 182 – Authors should include a brief description of the method, readers should be able to interpret the data without reading a separate manuscript.
Line 221 – What statistical software was used?
Figure 1 – Axis labels are way too small to read, and size needs to be increased on all text included in the figure. The bright yellow color is also very difficult to see and should be switched out.
Figure 2 – Same comment as figure 1.
